# A Regularized Weighted Smoothed *L*_0_ Norm Minimization Method for Underdetermined Blind Source Separation

**DOI:** 10.3390/s18124260

**Published:** 2018-12-04

**Authors:** Linyu Wang, Xiangjun Yin, Huihui Yue, Jianhong Xiang

**Affiliations:** College of Information and Communication Engineering, Harbin Engineering University, Harbin 150001, China; wanglinyu@hrbeu.edu.cn (L.W.); yinxiangjun@hrbeu.edu.cn (X.Y.); xiangjianhong@hrbeu.edu.cn (J.X.)

**Keywords:** image reconstruction, nullspace measurement matrix, regularized least squares problem, smoothed *L*_0_-norm, sparse signal recovery, UBSS, weighted function

## Abstract

Compressed sensing (CS) theory has attracted widespread attention in recent years and has been widely used in signal and image processing, such as underdetermined blind source separation (UBSS), magnetic resonance imaging (MRI), etc. As the main link of CS, the goal of sparse signal reconstruction is how to recover accurately and effectively the original signal from an underdetermined linear system of equations (ULSE). For this problem, we propose a new algorithm called the weighted regularized smoothed $L_{0}$-norm minimization algorithm (WReSL0). Under the framework of this algorithm, we have done three things: (1) proposed a new smoothed function called the compound inverse proportional function (CIPF); (2) proposed a new weighted function; and (3) a new regularization form is derived and constructed. In this algorithm, the weighted function and the new smoothed function are combined as the sparsity-promoting object, and a new regularization form is derived and constructed to enhance de-noising performance. Performance simulation experiments on both the real signal and real images show that the proposed WReSL0 algorithm outperforms other popular approaches, such as SL0, BPDN, NSL0, and L${}_{p}$-RLSand achieves better performances when it is used for UBSS.

## 1. Introduction

The problem that UBSS [1,2] needs to address is how to separate multiple signals from a small number of sensors. The essence of this problem is to solve the optimal solution of the undetermined linear system of equations (ULSE). Fortunately, as a new undersampling technique, compressed sensing (CS) [3,4,5] is an effective way to solve ULSE, which makes it possible to apply CS to UBSS.

The model of CS is shown in Figure 1. According to this figure, it can be see that CS boils down to the form,
(1)y=Φx+b,
where $\Phi =\left[\phi_{1},\phi_{2},\ldots ,\phi_{n}\right]\in R^{m\times n}$ is a sensing matrix with the condition of m≪n and $\phi_{i}\in R^{m},$
i=1,2,…,n, which can be further represented as Φ=ψφ, while ψ is a random matrix, and φ is the sparse basis matrix. $y\in R^{m}$ is the vector of measurements. Moreover, $b\in R^{m}$ denotes the additive noise.

To solve the ULSE in Equation (Equation 1), we try to recover the sparse signal x from the given {y,Φ} by CS. According to CS, this problem is transformed into solving the $L_{0}$-norm minimization problem.
(2)$$\left(P_{0}\right)\;\;\underset{x\in R^{n}}{\arg\;\min}{\left\|x\right\|}_{0},\;\;s.t.\;\|\Phi x-{y\|}_{2}^{2}\leq \epsilon .$$
where ϵ denotes error. This rather wonderful attempt is actually supported by a brilliant theory [6]. Based on this theory, in the noiseless case, it is proven that the sparsest solution is indeed a real signal when x is sufficiently sparse and Φ satisfies the restricted isometry property (RIP) [7]:(3)$$1-\delta_{K}\leq \frac{{\left\|\Phi x\right\|}_{2}^{2}}{{\left\|x\right\|}_{2}^{2}}\leq 1+\delta_{K},$$
where *K* is the sparsity of signal x and $\delta_{K}\in \left(0,1\right)$ is a constant. In Equation (Equation 2), the $L_{0}$-norm is nonsmooth, which leads an NP-hard problem. In practice, two alternative approaches are usually employed to solve the problem [8]:Greedy search by using the known sparsity as a constraint;The relaxation method for the $P_{0}$.

For greedy search, the main methods are based on greedy matching pursuit (GMP) algorithms, such as orthogonal matching pursuit (OMP) [9,10], stage-wise orthogonal matching pursuit (StOMP) [11], regularized orthogonal matching pursuit (ROMP) [12], compressive sampling matching pursuit (CoSaMP) [13], generalized orthogonal matching pursuit (GOMP) [14,15], and subspace pursuit (SP) [16,17] algorithms. The objective function of these algorithms is given by:(4)$$\underset{x\in R^{n}}{\arg\;\min}\frac{1}{2}{∥\Phi x-y∥}_{2}^{2},\;s.t.\;{∥x∥}_{0}\leq K.$$

As shown in the above equation, the features of GMP algorithms can be concluded as:Using sparsity as prior information;Using the least squares error as the iterative criterion.

The advantage of GMP algorithms is that the computational complexity is low, but the reconstruction accuracy is not high in the noise case.

At present, the relaxation method for $P_{0}$ is widely used. The relaxation method is mainly divided into two categories: the constraint-type algorithm and the regularization method. The constraint-type algorithm can also be divided into $L_{1}$-norm minimization methods and smoothed $L_{0}$-norm minimization methods. The representative algorithm of the former is the BPalgorithm [18], and the latter is the smoothed $L_{0}$-norm minimization (SL0) algorithm. For the SL0 algorithm, the objective function can be expressed as:(5)$$\begin{array}{c} \left(P_{F}\right)\;\;\underset{x\in R^{n}}{\arg\;\min}F_{\sigma }(x),\;\;s.t.\;\|\Phi x-{y\|}_{2}^{2}\leq \epsilon .\\ \lim_{\sigma \to 0}F_{\sigma }\left(x\right)=\lim_{\sigma \to 0}\sum_{i=1}^{n}f_{\sigma }\left(x_{i}\right)\approx {∥x∥}_{0}. \end{array}$$
where $F_{\sigma }(x)$ is a smoothed function, which approximates the $L_{0}$-norm when σ→0. Compared with $L_{1}$ or $L_{p}$, a small σ is selected to make the function close to $L_{0}$-norm [8]; therefore, $F_{\sigma }(x)$ are closer to the optimal solution.

Based on the idea of approximation, Mohimani used a Gauss function to approximate the $L_{0}$-norm [19], which is described as:(6)$$f_{\sigma }\left(x_{i}\right)=1-\exp(-\frac{x_{i}^{2}}{2\sigma^{2}}).$$

According to the equation, we can know:(7)$$f_{\sigma }\left(x_{i}\right)\approx \left\{\begin{array}{cc} 1 & \mathrm{if}\;x_{i}\gg \sigma \\ 0 & \mathrm{if}\;x_{i}\ll \sigma . \end{array}\right.$$
when σ is a small enough positive value, the Gauss function is almost equal to the $L_{0}$-norm. Furthermore, the Gauss function is differentiable and smoothed; hence, it can be optimized by optimization methods such as the gradient descent (GD) method. Zhao proposed another smoothed function: the hyperbolic tangent (tanh) [20],
(8)$$f_{\sigma }\left(x_{i}\right)=\frac{\exp\left(\frac{x_{i}^{2}}{2\sigma^{2}}\right)-\exp\left(-\frac{x_{i}^{2}}{2\sigma^{2}}\right)}{\exp\left(\frac{x_{i}^{2}}{2\sigma^{2}}\right)+\exp\left(-\frac{x_{i}^{2}}{2\sigma^{2}}\right)}.$$

This smoothed function makes a closer approximation to the $L_{0}$-norm than the Gauss function, as shown in [19], with the same σ; hence, it performs better in sparse signal recovery. Indeed, a large number of simulation experiments confirmed this view.

Another relaxation method is the regularization method. For CS, sparse signal recovery in the noise case is a very practical and unavoidable problem. Fortunately, the regularization method makes the solution of this problem possible [21,22]. The regularization method can be described as a “relaxation” approach that tries to solve the following unconstrained recovery problem:(9)$$\left(P_{\upsilon }\right)\underset{x\in R^{n}}{\arg\;\min}\frac{1}{2}{∥\Phi x-y∥}_{2}^{2}+\lambda \upsilon \left(x\right),$$
where λ>0 is the parameter that balances the trade-off between the deviation term ${∥\Phi x-y∥}_{2}^{2}$ and the sparsity regularizer υ(x). The sparse prior information is enforced via the regularizer υ(x), and a proper υ(x) is crucial to the success of the sparse signal recovery task: it should favor sparse solutions and make sure the problem $P_{\upsilon }$ can be solved efficiently in the meantime.

For regularization, various sparsity regularizers have been proposed as the relaxation of the $L_{0}$-norm. The most popular algorithms are the convex $L_{1}$-norm [22,23] and the nonconvex $L_{p}$-norm to the $p^{\mathrm{th}}$ power [24,25]. In the noiseless case, the $L_{1}$-norm is equivalent to the $L_{0}$-norm, and the $L_{1}$-norm is the only norm with sparsity and convexity. Hence, it can be optimized by convex optimization methods. However, according to [8], in the noisy case, the $L_{1}$-norm is not exactly equivalent to the $L_{0}$-norm, so the effect of promoting sparsity is not obvious. Compared to the $L_{1}$-norm, the nonconvex $L_{p}$-norm to the $p^{\mathrm{th}}$ power makes a closer approximation to the $L_{0}$-norm; therefore, $L_{p}$-norm minimization has a better sparse recovery performance [8].

In view of the above explanation, in this paper, a compound inverse proportional function (CIPF) function is proposed as a new smoothed function, and a new weighted function is proposed to promote sparsity. For the noise case, a new regularization form is derived and constructed to enhance de-noising performance. The experimental simulation verifies the superior performance of this algorithm in signal and image recovery, and it has achieved good results when applied to UBSS.

This paper is organized as follows: Section 2 introduces the main work of this paper. The steps of the ReRSL0algorithm and the selection of related parameters are described in Section 3. Experimental results are presented in Section 4 to evaluate the performance of our approach. Section 5 verifies the effect of the proposed weighted regularized smoothed $L_{0}$-norm minimization (WReSL0) algorithm in UBSS. Section 6 concludes this paper.

## 2. Main Work of This Paper

In this paper, based on the $P_{F}$ in Equation (Equation 9), we propose a new objective function, which is given by:(10)$$\underset{x\in R^{n}}{\arg\;\min}WH_{\sigma }(x),\;\;s.t.\;\|\Phi x-{y\|}_{2}^{2}\leq \epsilon .$$

According to this equation, We not only propose a smoothed function approximating the $L_{0}$-norm, but also propose a weighted function to promote sparsity. This section focuses on the relevant contents of $W={\left[w_{1},w_{2},\ldots w_{n}\right]}^{T}$ and $H_{\sigma }\left(x\right)$.

### 2.1. New Smoothed Function: CIPF

According to [26], some properties of the smoothed functions are summarized in the following:

**Property**: Let f:R→[−∞,+∞] and, define $f_{\sigma }\left(r\right)\approx f_{\sigma }\left(r/\sigma \right)$ for any σ>0. The function *f* has the property, if:(a)*f* is real analytic on $\left(r_{0},\infty \right)$ for some $r_{0}$;(b)∀r≥0, $f^{{}^{\prime \prime }}\left(r\right)\geq -\epsilon_{0}$, where $\epsilon_{0}>0$ is some constant;(c)*f* is convex on R;(d)f(r)=0↔r=0;(e)$\lim_{r\to +\infty }f\left(r\right)=1$.

It follows immediately from Property that $\left\{f_{\sigma }\left(r\right)\right\}$ converges to the $L_{0}$-norm as $\sigma \to 0^{+}$, i.e.,
(11)$$\lim_{\sigma \to 0^{+}}f_{\sigma }\left(r\right)=\left\{\begin{array}{cc} 0 & \mathrm{if}\;\;r=0\\ 1 & \;\;\mathrm{otherwise}. \end{array}\right.$$

Based on Property, this paper proposes a new smoothed function model called CIPF, which satisfies Property and better approximates the $L_{0}$-norm. The smoothed function model is given as:(12)$$f_{\sigma }\left(r\right)=1-\frac{\sigma^{2}}{\alpha r^{2}+\sigma^{2}}.$$

In Equation (Equation 12), α denotes a regularization factor, which is a large constant. By experiments, the factor α is determined to be 10, which is a good result of the simulation. σ represents a smoothed factor, and when it is smaller, it will make the proposed model closer to the $L_{0}$-norm. Obviously, $\lim_{\sigma \to 0}f_{\sigma }\left(r\right)=\left\{\begin{array}{cc} 0, & r=0\\ 1, & r\neq 0 \end{array}\right.$ or approximately $f_{\sigma }\left(r\right)\approx \left\{\begin{array}{cc} 0, & |\;r|\ll \sigma \\ 1, & |\;r|\gg \sigma \end{array}\right.$ is satisfied. Let:(13)$$H_{\sigma }\left(x\right)=\sum_{i=1}^{n}f_{\sigma }\left(x_{i}\right)=n-\sum_{i=1}^{n}\frac{\sigma^{2}}{\alpha x_{i}^{2}+\sigma^{2}}$$
where $H_{\sigma }\left(x\right)\approx {\left\|x\right\|}_{0}$ for small values of σ, and the approximation tends to equality when σ→0.

Figure 2 shows the effect of the CIPF model approximating the *L*_0_-norm. Obviously, the CIPF model makes a better approximation.

In conclusion, the merits of the CIPF model can be summarized as follows:It closely approximates the $L_{0}$-norm;It is simpler in form than that in the Gauss and tanh function models.

These merits make it possible to reduce the computational complexity on the premise of ensuring the accuracy of sparse signal reconstruction, which is of practical significance for sparse signal reconstruction.

### 2.2. New Weighted Function

Candès et al. [27] proposed the weighted $L_{1}$-norm minimization method, which employs the weighted norm to enhance the sparsity of the solution. They provided an analytical result of the improvement in the sparsity recovery by incorporating the weighted function with the objective function. Pant et al. [28] applied another weighted smoothed $L_{0}$-norm minimization method, which uses a similar weighted function to promote sparsity. The weighted function can be summarized as follows:Candès et al.: $w_{i}=\left\{\begin{array}{cc} \frac{1}{\left|x_{i}\right|} & x_{i}\neq 0\\ \infty & x_{i}=0 \end{array}\right.$;Pant et al.: $w_{i}=\frac{1}{|x_{i}|+\zeta }$, ζ is a small enough positive constant.

From the two weighted functions, we can find a phenomenon: a large signal entry $x_{i}$ is weighted with a small $w_{i}$; on the contrary, a small signal entry $x_{i}$ is weighted with a large value $w_{i}$. By analysis, the large $w_{i}$ forces the solution x to concentrate on the indices where $w_{i}$ is small, and by construction, these correspond precisely to the indices where x is nonzero.

Combined with the above idea, we propose a new weighted function, which is given by:(14)$$w_{i}=e^{-\frac{|x_{i}|}{\sigma }},\;\;s.t.\;i=1,2,\ldots ,n.$$

As for Candès et al., when the signal entry is zero or close to zero, the value of $w_{i}$ will be very large, which is not suitable for computation by a computer. Although Pant et al. noticed the problem and improved the weighted function to avoid it, the constant ζ depends on experience. Actually, the proposed weighted function can avoid the two problems. Moreover our weighted function can be satisfied with the phenomenon. When the small signal entry $x_{i}$ can be weighted with a large $w_{i}$ and a large signal entry $x_{i}$ can be weighted with a small $w_{i}$, this can make the large signal entry and small signal entry closer. In this way, the direction of optimization can be kept as consistent as possible, and the optimization process tends to be more optimal. Therefore, the proposed weighted function can have a better effect.

## 3. New Algorithm for CS: WReSL0

### 3.1. WReSL0 Algorithm and Its Steps

Here, in order to analyze the problem more clearly, we rewrite Equation (Equation 10) as follows:$$\underset{x\in R^{n}}{\arg\;\min}WH_{\sigma }\left(x\right),\;\;s.t.\;\|\Phi x-{y\|}_{2}^{2}\leq \epsilon .$$
where $H_{\sigma }\left(x\right)=I-\frac{\sigma^{2}}{\alpha x^{2}+\sigma^{2}}$ ($I\in R^{N}$ is a unit vector) is a differentiable smoothed accumulated function. The weighted function $W=e^{-}\frac{\left|x\right|}{\sigma }$. Therefore, we can obtain the gradient of CIPF, which is written as:(15)$$G=\frac{\partial H_{\sigma }\left(x\right)}{\partial x}=\frac{2\alpha \sigma^{2}x}{{\left(\alpha x^{2}+\sigma^{2}\right)}^{2}}$$

According to Equation (Equation 15), as in [28], we can obtain:(16)$$\mathrm{WG}={\left(e^{-}\frac{\left|x\right|}{\sigma }\right)}^{T}\frac{2\alpha \sigma^{2}x}{{\left(\alpha x^{2}+\sigma^{2}\right)}^{2}}$$

Solving the problem of ULSE is to solve the optimization problem in Equation (Equation 10). As for this problem, there are many methods, such as split Bregman methods [29,30,31], FISTA [32], alternating direction methods [33], gradient descent (GD) [34], etc. In order to reduce the computational complexity, this paper adopts the GD method to optimize the proposed objective function.

Given σ, a small target value $\sigma_{\min}$, and a sufficiently large initial value $\sigma_{\max}$, after referring to the annealing mechanism in simulated annealing [35], this paper proposes a monotonically-decreasing sequence $\left\{\sigma_{t}|t=2,3,\ldots ,T\right\}$, which is generated as:(17)$$\sigma_{t}=\sigma_{\max}\theta^{-\gamma (t-1)},\;\;s.t.\;t=1,2,3,\ldots ,T.$$
where $\gamma =\frac{\log_{\theta }\left(\sigma_{\max}/\sigma_{\min}\right)}{T-1}$, θ is a constant that is larger than one, and *T* is the maximum number of iterations. Using such a monotonically-decreasing sequence can avoid the case of too small of a σ leading to the local optimum.

Similar to SL0, WReSL0 also consists of two nested iterations: the external loop, which begins with a sufficiently large value of σ, i.e, $\sigma_{\max}$, responsible for the gradually decreasing strategy in Equation (Equation 17), and the internal loop, which for each value of σ, finds the maximizer of $H_{\sigma }\left(x\right)$ on $\left\{x|\|Ax-y{\|}_{2}\leq \epsilon \right\}$.

According to the GD algorithm, the internal loop consists of the gradient descent step, which is given by:(18)$$\hat{x}=x+\mu d,$$
where d=g and μ denotes a step size factor. This part is similar to SL0, followed by solving the problem:(19)$$\underset{x^{*}\in R^{n}}{\arg\;\min}\|x^{*}-\hat{x}{\|}_{2}^{2},\;\;s.t.\|\Phi x^{*}-y{\|}_{2}^{2}\leq \epsilon$$
where $x^{*}$ denotes the optimal solution. By regularization, this form can be converted to another form as follows,
(20)$$\underset{x^{*}\in R^{n}}{\arg\;\min}\|x^{*}-\hat{x}{\|}_{2}^{2}+\lambda \|\Phi x^{*}-y{\|}_{2}^{2}.$$
where λ is the regularization parameter, which is adapted to balance the fit of the solution to the data *y* and the approximation of the solution to the maximizer of $H_{\sigma }\left(x\right)$. Weighted least squares (WLS) can be used to solve this problem, and the solution is:(21)$$x^{*}={\left[\begin{array}{ccc} {\left[\begin{array}{c} I_{n}\\ \Phi \end{array}\right]}^{H} & \left[\begin{array}{cc} I_{n} & 0\\ 0 & \lambda I_{m} \end{array}\right] & \left[\begin{array}{c} I_{n}\\ \Phi \end{array}\right] \end{array}\right]}^{-1}{\left[\begin{array}{c} I_{n}\\ \Phi \end{array}\right]}^{H}\left[\begin{array}{cc} I_{n} & 0\\ 0 & \lambda I_{m} \end{array}\right]\left[\begin{array}{c} \hat{x}\\ y \end{array}\right].$$

By calculation, Equation (Equation 21) is equivalent to:(22)$$x^{*}={\left(I_{n}+\lambda \Phi^{H}\Phi \right)}^{-1}\left(\hat{x}+\lambda \Phi^{H}y\right)$$
where $I_{n}$ and $I_{m}$ are both identity matrices of size n×n and m×m, respectively. Therefore, we can obtain:$$\begin{array}{cc} x^{*}-\hat{x} & ={\left(I_{n}+\lambda \Phi^{H}\Phi \right)}^{-1}\left(\hat{x}+\lambda \Phi^{H}y\right)-\hat{x}\\ & ={\left(I_{n}+\lambda \Phi^{H}\Phi \right)}^{-1}\left(\hat{x}+\lambda \Phi^{H}y-\left(I_{n}+\lambda \Phi^{H}\Phi \right)\hat{x}\right)\\ & ={\left(I_{n}+\lambda \Phi^{H}\Phi \right)}^{-1}\left(\hat{x}+\lambda \Phi^{H}y-\hat{x}-\lambda \Phi^{H}\Phi \hat{x}\right)\\ & =-{\left(\lambda^{-1}I_{n}+\Phi^{H}\Phi \right)}^{-1}\Phi^{H}\left(\Phi \hat{x}-y\right) \end{array}$$

According to the above analysis and derivation, we can get:(23)$$x^{*}=\hat{x}-{\left(\lambda^{-1}I_{n}+\Phi^{H}\Phi \right)}^{-1}\Phi^{H}\left(\Phi \hat{x}-y\right)$$

The initial value of the internal loop is the maximizer of $H_{\sigma }\left(x\right)$ obtained for $\sigma_{\max}$. To increase the speed, the internal loop is repeated a fixed and small number of times (L). In other words, we do not wait for the GD method to converge in the internal loop.

According to the explanation above, we can conclude the steps of the proposed WReSL0 algorithm, which are given in Table 1. As for σ, it can be shown that function $H_{\sigma }\left(x\right)$ remains convex in the region where the largest magnitude of the component of x is less than σ. As the algorithm starts at the original value $x^{\left(0\right)}=\Phi^{H}(\Phi \Phi^{H}{)}^{-1}y$, the above choice of $\sigma_{1}$ ensures that the optimization starts in a convex region. This greatly facilitates the convergence of the WReSL0 algorithm.

### 3.2. Selection of Parameters

The selection of parameters μ and σ will affect the performance of the WReSL0 algorithm; thus, this paper discusses the selection of these two above parameters in this section.

#### 3.2.1. Selection of Parameter μ

According to the algorithm, each iteration consists of a descent step $x_{i}\leftarrow x_{i}-\mu \left(e^{-}\frac{|x_{i}|}{\sigma }\right)\frac{2\alpha \sigma^{2}x_{i}}{{\left(\alpha {x_{i}}^{2}+\sigma^{2}\right)}^{2}},1\leq i\leq n$, followed by a projection step. If for some values of *i*, we have $|x_{i}|\gg \sigma$, then the algorithm does not change the value of $x_{i}$ in that descent step; however, it might be changed in the projection step. If we are looking for a suitably large μ, a suitable choice is to make the algorithm force all those values of x satisfying $|x_{i}|≲\sigma$ toward zero. Therefore, we can get:(24)$$x_{i}-\mu \left(e^{-}\frac{|x_{i}|}{\sigma }\right)\frac{2\alpha \sigma^{2}x_{i}}{{\left(\alpha {x_{i}}^{2}+\sigma^{2}\right)}^{2}}\approx 0$$
and:(25)$$\left(e^{-}\frac{|x_{i}|}{\sigma }\right)\overset{x_{i}\to 0}{\to }1$$

Combining Equations (Equation 24) and (Equation 25), we can further obtain:(26)$$x_{i}-\mu \frac{2\alpha \sigma^{2}x_{i}}{{\left(\alpha {x_{i}}^{2}+\sigma^{2}\right)}^{2}}\approx 0$$

By calculation, we can obtain:(27)$$\mu \approx \frac{{\left(\alpha {x_{i}}^{2}+\sigma^{2}\right)}^{2}}{2\alpha \sigma^{2}}\overset{x_{i}\to 0}{\to }\frac{\sigma^{2}}{2\alpha }$$

According to the above derivation, we have come to the conclusion that $\mu \approx \frac{\sigma^{2}}{2\alpha }$. Therefore, we can set $\mu =\frac{\sigma^{2}}{2\alpha }$.

#### 3.2.2. Selection of Parameter σ

According to Equation (Equation 17), the descending sequence of σ is generated by $\sigma_{t}=\sigma_{\max}{\left(\frac{\sigma_{\min}}{\sigma_{\max}}\right)}^{\frac{t-1}{T-1}}$ (it is obtained through simplification of Equation (Equation 17)). Parameter $\sigma_{\min}$ and parameter $\sigma_{\max}$ should be appropriately selected. The selection of $\sigma_{\min}$ and $\sigma_{\max}$ is discussed below.

For the initial value of σ, i.e., $\sigma_{\max}$, here, let $\tilde{x}=\max\left\{|x|\right\}$; suppose there is a constant *b*, in order to make the algorithm converge quickly; let parameter $\sigma_{\max}$ satisfy:(28)$$H_{\sigma }\left(\tilde{x}\right)=1-\frac{\sigma_{\max}^{2}}{\alpha {\tilde{x}}^{2}+\sigma_{\max}^{2}}\leq b\Rightarrow \sigma_{\max}\geq \left(\sqrt{\frac{1-b}{b}\alpha }\right)\tilde{x}.$$

From the equation, we can see that constant *b* satisfies $\frac{1-b}{b}\geq 0$; thus 0<b≤1, and here, we define constant *b* as 0.5. Hence, $\sigma_{\max}=\sqrt{\alpha }\max\left\{|x|\right\}$.

For the final value $\sigma_{\min}$, when $\sigma_{\min}\to 0$, $H_{\sigma_{\min}}\left(x\right)\to {\left\|x\right\|}_{0}$. That is, the smaller $\sigma_{\min}$, the more $H_{\sigma_{\min}}\left(x\right)$ can reflect the sparsity of signal x, but at the same time, it is also more sensitive to noise; therefore, the value $\sigma_{\min}$ should not be too small. Combining [19], we choose $\sigma_{\min}=0.01$.

## 4. Performance Simulation and Analysis

The numerical simulation platform is MATLAB 2017b, which is installed on a computer with a Windows 10, 64-bit operating system. The CPU of the simulation computer is the Intel (R) Core (TM) i5-3230M, and the frequency is 2.6 GHz. In this section, the performance of the WReSL0 algorithm is verified by signal and image recovery in the noise case.

Here, some state-of-the-art algorithms are selected for comparison. The parameters are selected to obtain the best performance for each algorithm: for the BPDNalgorithm [36], the regularization parameter $\lambda =\sigma_{N}\sqrt{2\log(n)}$; for the SL0 algorithm [19], the initial value of smoothed factor $\delta_{\max}=2\max\left\{|x|\right\}$, the final value of smoothed factor $\delta_{\min}=0.01$, scale factor is set as step size L=5, and the attenuation factor ρ=0.8; for the NSL0algorithm [20], the initial value of smoothed factor $\delta_{\max}=4\max\left\{x\right\}$, the final value of smoothed factor $\delta_{\min}=0.01$, the step size L=10, and the attenuation factor ρ=0.8; for L${}_{p}$-RLSalgorithm [24], the number of iterations T=80, the norm initial value $p_{1}=1$, the norm final value $p_{T}=0.1$, the initial value of regularization factor $\epsilon_{1}=1$, the final value of regularization factor $\epsilon_{T}=0.01$, and the algorithm termination threshold $E_{t}=10^{-25}$; for the WReSL0 algorithm, the initial value of smoothed factor $\sigma_{\max}=\sqrt{c}\max\left\{|x|\right\}$, the final value of smoothed factor $\sigma_{\min}=0.01$, the iterations T=30, the step size L=5, and the regularization parameter λ=0.1. All experiments are based on 100 trials.

### 4.1. Signal Recovery Performance in the Noise Case

In this part, we discuss signal recovery performance in the noise case. We add noise b to the measurement vector y; moreover, $b=\delta_{N}\Omega$, Ω is randomly formed and follows the Gaussian distribution of N(0,1). For signal recovery under noise conditions, we evaluate the performance of algorithms by the normalized mean squared error (NMSE) and the CPU running time (CRT). NMSE is defined as $\|x-\hat{x}{\|}_{2}/{\left\|x\right\|}_{2}$. CRT is measured with tic and toc. In order to analyze the de-noising performance of the WReSL0 algorithm in context closer to the real situation, we constructed a certain signal as an experimental object in the experiments in this section. The signal is given by:(29)$$\left\{\begin{array}{c} x_{1}=\alpha_{1}\sin\left(2\pi f_{1}T_{s}t\right)\\ x_{2}=\beta_{1}\cos\left(2\pi f_{2}T_{s}t\right)\\ x_{3}=\alpha_{2}\sin\left(2\pi f_{3}T_{s}t\right)\\ x_{4}=\beta_{2}\cos\left(2\pi f_{4}T_{s}t\right)\\ X=x_{1}+x_{2}+x_{3}+x_{4} \end{array}\right.$$
where $\alpha_{1}=0.2$, $\alpha_{2}=0.1$, $\beta_{1}=0.3$, and $\beta_{2}=0.4$. $f_{1}=50$ Hz; $f_{2}=100$ Hz; $f_{3}=200$ Hz; and $f_{4}=300$ Hz. Here, t is a sequence with t=[1,2,3,…,n], and $T_{s}$ is sampling interval with the value of $\frac{1}{f_{s}}$. $f_{s}$ is the sampling frequency with the value of 800 Hz. The object that needs to be reconstructed can be expressed as:(30)$$y=\Phi x+\delta_{N}\Omega .$$
where $x\in R^{n}$ is a sparse signal in the frequency domain, and it is the Fourier transform expression of X, $y\in R^{m}$. Here, let n=128, m=64. Moreover, Φ can be represented as Φ=ψφ; here, ψ is a randn matrix generated by a Gaussian distribution, and φ is a sparse basis matrix generated by Fourier transform. Here, φ can be given by Fourier $I_{n\times n}\;$, and $I_{n\times n}$ is a unit matrix. This target signal X is sparse in Fourier space; hence, the signal X can be recovered from given {y,Φ} by CS recovery methods.

Figure 3 shows the signal recovery effect. Obviously, BPDN and SL0 do not perform well, while NSL0, L${}_{p}$-RLS and the proposed WReSL0 perform quite well. This verifies that the regularization mechanism has a good de-noising effect. Figure 4 shows the frequency spectrum of the recovered signal by the selected algorithms. The spectrum of the signal recovered by our proposed WReSL0 algorithm is almost the same as the original signal, while other algorithms fail to achieve this effect.

Table 2 shows the CRT of all algorithms. The *n* changes according to a given sequence [170,220,270,320,370,420,470,520]. From the table, for any *n*, SL0 has the shortest computation time, followed by WReSL0, NSL0, and L${}_{p}$-RLS, and BPDN has the longest computation time. The BPDN algorithm is generally implemented by the quadratic programming method, and the computational complexity of this method is very high, thus resulting in a large increase in the overall computation time of the algorithm. Furthermore, in L${}_{p}$-RLS, the iterative process adopts the conjugate gradient method with high complexity, while NSL0 and WReSL0 do not. Compared with NSL0, WReSL0 is more prominent in the decrease of computation time.

The performance of each algorithm under different noise intensities is shown in Figure 5. When $\delta_{N}=0$, SL0 outperforms other algorithms, but with the increase of $\delta_{N}$, the effect of SL0 becomes worse and worse. This result further illustrates that the traditional constrained sparse recovery algorithm does not have the performance of anti-noising. For BPDN, NSL0, L${}_{p}$-RLS, and WReSL0, they all applied the regularization mechanism, and they are indeed superior to SL0 in the noise case. Therefore, the proposed WReSL0 in this paper has the best de-noising performance.

### 4.2. Image Recovery Performance in the Noise Case

Real images are considered to be approximately sparse under some proper basis, such as the DCT basis, DWT basis, etc. Here, we choose the DWT basis to recover these images. We compare the recovery performances based on the four real images in Figure 6: boat, Barbara, peppers, and Lena. The size of these images is 256×256; the compression ratio (CR; defined as m/n) is 0.5; and the noise $\delta_{N}$ equals 0.01. We still choose SL0, BPDN, NSL0, and L${}_{p}$-RLS to make comparisons. For image recovery, the object of image processing is given by:(31)Y=ΦX+B

Here, Y, X, B are matrices, and among these, $Y,B\in R^{m\times n}$, $X\in R^{n\times n}$. In order to meet the basic requirements of CS, we perform the following processing:(32)$$Y_{i}=\Phi X_{i}+B_{i}\;\;s.t.\;i=1,2,\ldots ,n.$$
where $Y_{i}$, $X_{i}$, $B_{i}$ are the column vectors of Y, X, B, respectively. $B_{i}=\delta_{N}\Omega$, Ω obeys the Gaussian distribution N(0,1).

To perform image recovery, we valuate it by the peak signal to noise ratio (PSNR) and the structural similarity index (SSIM). PSNR is defined as:(33)$$\mathrm{PSNR}=10\log(255^{2}/\mathrm{MSE})$$
where $\mathrm{MSE}=\|x-\hat{x}{\|}_{2}^{2}$, and SSIM is defined as:(34)$$\mathrm{SSIM}\left(p,q\right)=\frac{\left(2\mu_{p}+\mu_{q}+c_{1}\right)\left(2\sigma_{pq}+c_{2}\right)}{\left(\mu_{p}^{2}+\mu_{q}^{2}+c_{1}\right)\left(\sigma_{p}^{2}+\sigma_{q}^{2}+c_{2}\right)}.$$

Among these, $\mu_{p}$ is the mean of image *p*, $\mu_{q}$ is the mean of image *q*, $\sigma_{p}$ is the variance of image *p*, $\sigma_{q}$ is the variance of image *q*, and $\sigma_{pq}$ is the covariance between image *p* and image *q*. Parameters $c_{1}=z_{1}L$ and $c_{2}=z_{2}L$, for which $z_{1}=0.01,z_{2}=0.03$, and *L* is the dynamic range of pixel values. The range of SSIM is [−1,1], and when these two images are the same, SSIM equals one.

Figure 7 shows the recovery effect of boat and Barbara with noise intensity $\delta_{N}=0.01$. For boat and Barbara, the recovered images by SL0 and BPDN have obvious water ripples, while recovered images by other algorithms have no such water ripples. Similarly, for peppers and Lena, the recovered images by SL0 and BPDN are blurred compared with the recovered images by other algorithms. The NSL0, L${}_{p}$-RLS, and WReSL0 algorithms are also effective at noisy image recovery. For the NSL0, L${}_{p}$-RLS, and WReSL0 algorithms, their recovery effects are very similar. In order to further analyze the advantages and disadvantages of the algorithms, we analyze the PSNR and SSIM of the images recovered by these algorithms, and the results are shown in Table 3 and Table 4. By observation and analysis, L${}_{p}$-RLS performs better than NSL0, and at the same time, WReSL0 outperforms L${}_{p}$-RLS. Hence, the WReSL0 proposed by this paper is superior to the other selected algorithms in image processing.

## 5. Application in Underdetermined Blind Source Separation

The problem of UBSS stems from cocktail reception, which is shown in Figure 8. Suppose the source signal matrix $S\left(t\right)={\left[s_{1}\left(t\right),s_{2}\left(t\right),\ldots ,s_{m}\left(t\right)\right]}^{T}$, the mixed matrix (Sensors) A is m×n (m≪n) matrix, the Gaussian noise $G\left(t\right)={\left[g_{1}\left(t\right),g_{2}\left(t\right),\ldots ,g_{m}\left(t\right)\right]}^{T}$ is generated by Gaussian distribution, and the observed mixed signal matrix $X\left(t\right)={\left[x_{1}\left(t\right),x_{2}\left(t\right),\ldots ,x_{n}\left(t\right)\right]}^{T}$; therefore, the general mathematical models of UBSS can be summarized as:(35)X(t)=AS(t)+G(t)

In fact, each signal has *L* data collected; therefore, $X\in R^{m\times L}$, $A\in R^{m\times n}\;\left(m\ll n\right)$, $S\left(t\right)\in R^{n\times L}$, and $G\in R^{m\times L}$, and G can be represented as $\delta_{N}W$ (W obeys N(0,1)). The purpose of UBSS is to use the mixed signal matrix x(t) to estimate the sof the source signal matrix s(t). In fact, this is the process of solving the underdetermined linear system of equations (ULSE). For this problem, we can use the two-step method to solve it, which is shown in Figure 9.

From Figure 9, firstly, we get the mixed matrix by the clustering method and then use CS technology to separate the signal, so as to restore the original signal.

### 5.1. Process Analysis of CS Applied to UBSS

#### 5.1.1. Solving the Mixed Matrix by the Potential Function Method

In this section, we choose the potential function method to solve the mixed matrix A. To verify the performance of the proposed WReSL0 algorithm better, we choose four simulated signals and four real images to organize experiments in this section.

Suppose there are four source signals, which are:(36)$$\left\{\begin{array}{c} s_{1}\left(t\right)=5\sin\left(2\pi f_{1}t\right)\\ s_{2}\left(t\right)=5\sin\left(2\pi f_{2}t\right)\\ s_{3}\left(t\right)=5\sin\left(2\pi f_{3}t\right)\\ s_{4}\left(t\right)=5\sin\left(2\pi f_{4}t\right)\\ S={\left[s_{1}\left(t\right),s_{2}\left(t\right),s_{3}\left(t\right),s_{4}\left(t\right)\right]}^{T} \end{array}\right.$$
where $f_{1}=310$ Hz, $f_{2}=210$ Hz, $f_{3}=110$ Hz, and $f_{4}=10$ Hz. The length of each source signal $s_{i}\;\left(i=1,2,3,4\right)$ is 1024, and the sample frequency is 1024 Hz. These four signals are shown in Figure 10.

The four source images are the classic standard test images: boat, Barbara, peppers, and Lena, which are in Figure 6.

Suppose there are two sensors that receive signals and another two sensors that receive images. Mixed matrices A and B are set as:(37)$$\begin{array}{c} A=\left[\begin{array}{c} A_{1}\\ A_{2} \end{array}\right]=\left[\begin{array}{cccc} 0.9930 & 0.9941 & 0.1092 & 0.9304\\ 0.2116 & 0.0757 & 0.9647 & 0.3837 \end{array}\right]\\ B=\left[\begin{array}{c} B_{1}\\ B_{2} \end{array}\right]=\left[\begin{array}{cccc} 0.9354 & 0.9877 & -0.6730 & 0.1097\\ 0.3535 & 0.07846 & 0.7396 & 0.9940 \end{array}\right] \end{array}$$

By this mixed matrix and added Gaussian noise ($\delta_{N}=0.1$), we can get the two mixed signals, which are shown in Figure 11, and the two mixed images, which are shown in Figure 12. Then, we can get the estimated mixed matrix $\hat{A}$ and $\hat{B}$ by clustering by the potential function method [37]. As shown in Figure 13, the potential function method can cluster well. By clustering, we get the estimated values of A and B, as follows:(38)$$\begin{array}{c} \hat{A}=\left[\begin{array}{c} {\hat{A}}_{1}\\ {\hat{A}}_{2} \end{array}\right]=\left[\begin{array}{cccc} 0.9792 & 0.9969 & 0.1097 & 0.9239\\ 0.2028 & 0.0785 & 0.9940 & 0.3827 \end{array}\right]\\ \hat{B}=\left[\begin{array}{c} {\hat{B}}_{1}\\ {\hat{B}}_{2} \end{array}\right]=\left[\begin{array}{cccc} 0.9478 & 0.9431 & -0.6483 & 0.1130\\ 0.3476 & 0.0765 & 0.7075 & 0.9979 \end{array}\right] \end{array}$$

By calculation, the error of solving the mixed matrix is $\frac{\|A-\hat{A}|{|}_{F}}{\left\|A\right|{|}_{F}}\times 100\%=1.763\%$ and $\frac{\|B-\hat{B}|{|}_{F}}{\left\|B\right|{|}_{F}}\times 100\%=3.64\%$. This error range is much smaller than the classical k-means and fuzzy c-means, thus laying a foundation for the reconstruction of compressed sensing.

#### 5.1.2. Using CS to Separate Source Signals

The next problem is to get S(t) from known A(t) and X(t). Here, we solve this problem by CS. The solution process is similar to the image reconstruction process. The difference is that the sparse basis used here is the Fourier basis. Then, we apply the proposed RWeSL0 algorithm to this process. First, we transform the obtained x(t) into column vectors:(39)$$x\left(t\right)={\left[x_{1}\left(t\right),x_{2}\left(t\right)\right]}^{T}\Rightarrow \tilde{x}\left(t\right)=\left[\begin{array}{c} x_{1}\left(t\right)\\ x_{2}\left(t\right) \end{array}\right]$$

Then, we use the Fourier (for the sparse signal) or DWT (for the image) basis for sparse representation and extend the matrix and the valuated mixed matrix to obtain the sensing matrix.
(40)$$\begin{array}{c} \tilde{A}=\hat{A}⊗I_{L\times L},\;\;or\;\;\tilde{B}=\hat{B}⊗I_{L\times L}\\ \Psi =Fourier\left(I_{L\times L}\right)/\sqrt{L},\;\;or\;\;\Psi =DWT\left(I_{L\times L}\right)/\sqrt{L}\\ \tilde{\Psi }=\left[\begin{array}{cccc} \Psi & 0 & \ldots & 0\\ 0 & \Psi & \ldots & \ldots \\ \ldots & \ldots & \ldots & 0\\ 0 & \ldots & 0 & \Psi \end{array}\right]\\ \Phi =\tilde{A}\tilde{\Psi },\;\;or\;\;\Phi =\tilde{B}\tilde{\Psi } \end{array}$$

For this equation, ⊗ denotes the Kronecker product sign, Fourier(·) represents the Fourier transform, and DWT represents the discrete wavelet transform. Therefore, the CS-UBSS model can be described as:(41)$$\begin{array}{cc} \hat{X}\left(t\right) & =\tilde{A}\left(t\right)S\left(t\right)+G\left(t\right)\\ & =\tilde{A}\left(t\right)\tilde{\Psi }\Theta \left(t\right)+G\left(t\right)\\ & =\Phi \Theta (t)+G(t)\\ or & \\ \hat{X}\left(t\right) & =\tilde{B}\left(t\right)S\left(t\right)+G\left(t\right)\\ & =\tilde{B}\left(t\right)\tilde{\Psi }\Theta \left(t\right)+G\left(t\right)\\ & =\Phi \Theta (t)+G(t) \end{array}$$
where Θ is the Fourier transform or DWT of S(t), so Θ is a sparse signal. As for UBSS in the images, firstly, each image matrix needs to be transformed into a row vector, then the four row vectors form a matrix S(t). At the same time, the sparse basis in Equation (Equation 40) needs to be replaced by DWT.

Then, we can recover the source signal by CS. In summary, the above can be described as the flowchart in Figure 14.

### 5.2. Performance Analysis of the WReSL0 Algorithm Applied to UBSS

#### 5.2.1. The Effect of the WReSL0 Algorithm Applied to UBSS

In this section, we evaluate the effect of the WReSL0 algorithm applied to UBSS by the separation of signals and spectrum analysis.

The effect of the separation of signals is shown in Figure 15: the source signals are well separated, and the separation signals and the original signals are very similar. Figure 16 displays the error between the original source signal and the recovered source signal. It indicates that the error between the original source signal and the recovered source signal is fairly small, and the WReSL0 algorithm can better deal with the problem of UBSS. In addition, We get the time-frequency diagram of the restored signal by short-time Fourier transform. Figure 17 is the time-frequency diagram. From this figure, we find that each signal has the same frequency as the original signal, and it also validates the rationality of the proposed algorithm for UBSS.

#### 5.2.2. Performance Comparisons of the Selected Algorithms

Here, we use the SL0, NSL0, and L${}_{p}$-RLS algorithms and the classical shortest path method (SPM) [38] to make a comparison in different noise cases. In order to analyze the situation of signal recovery clearly, we apply average SNR (ASNR) (for the signal) and average peak SNR (APSNR) (for the image) to evaluate. Let the original source signal be $s_{i}$ and the recovered source signal be ${\hat{s}}_{i}$, so ANSR is defined as:(42)$$\begin{array}{c} \mathrm{ASNR}=\frac{1}{n}\sum_{i=1}^{n}{\mathrm{SNR}}_{i}\\ {\mathrm{SNR}}_{i}=20\log\frac{\|{\hat{s}}_{i}-s_{i}|{|}_{2}}{\|s_{i}|{|}_{2},} \end{array}$$
and PSNR is defined as:(43)$$\begin{array}{c} \mathrm{APSNR}=\frac{1}{n}\sum_{i=1}^{n}{\mathrm{PSNR}}_{i}\\ {\mathrm{PSNR}}_{i}=10\log\frac{255^{2}\times M\times N}{\|{\hat{s}}_{i}-s_{i}|{|}_{2}} \end{array}$$
where *M* and *N* are the width and height of the image.

The ASNR comparisons are shown in Table 5. From the table, we can see that ASNR attenuates sharply when $\delta_{N}$ increases from 0.15–0.2. The reason is that the error of the valuated mixed matrix $\hat{A}$ increases obviously, which leads those CS recovery algorithms to perform poorly. In fact, from this table, our proposed RWeSL0 algorithm performs well when $\delta_{N}$ is less than 0.15, and when $\delta_{N}$ is greater than 0.15, the L${}_{p}$-RLS algorithm performs best, followed by our proposed RWeSL0 algorithm.

The APSNR comparisons are shown in Table 6. In this table, It is clear that APSNR is not high, and it drops greatly when $\delta_{N}$ increases from 0.15–0.2. From Figure 18, we can see that these separated images seem to be enveloped in mist, which leads to a low APSNR. Therefore, we will try our best to improve this problem in the future.

In summary, the CS technique can be used in UBSS and performs well especially for the signal recovery. Our proposed WReSL0 algorithm can perform well in UBSS for the signal recoverywhen the noise is small; and regarding image recovery, we will develop this in the future.

## 6. Conclusions

In this paper, we propose the WReSL0 algorithm to recover the sparse signal from given {y,Φ} in the noise case. The WReSL0 algorithm is constructed under the GD method, in which the update process of **x** in the inner loop adopts the regularization mechanism to enhance the de-noising performance. As a key part of the WReSL0 algorithm, a weighted smoothed function $W^{T}H_{\sigma }\left(x\right)$ is proposed to promote sparsity and provide the guarantee of robust and accurate signal recovery. Furthermore, We deduced the value of μ and the initial value $\sigma_{\max}$ to ensure the optimization performance of the algorithm. Performance simulation experiments on both real signals and real images show that the proposed WReSL0 algorithm performs better than the $L_{1}$ or $L_{p}$ regularization methods and the classical $L_{0}$ regularization methods. Finally, we apply the proposed WReSL0 algorithm to solve the problem of UBSS and also make comparisons with the classical SPM, SL0, NSL0, and Lp-RLS algorithms. Experiments show that this algorithm has some advanced performance. In addition, we would also like to apply the the proposed algorithm to other CS applications such as the RPCA [39], SAR imaging [40], and other de-noising methods [41].

## Figures and Tables

**Figure 1 sensors-18-04260-f001:**
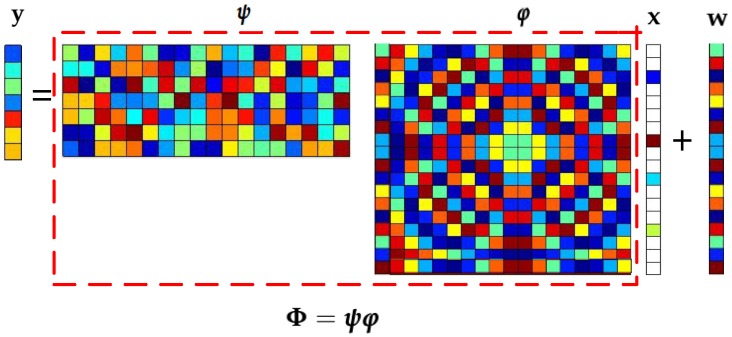
Frame of compressed sensing (CS).

**Figure 2 sensors-18-04260-f002:**
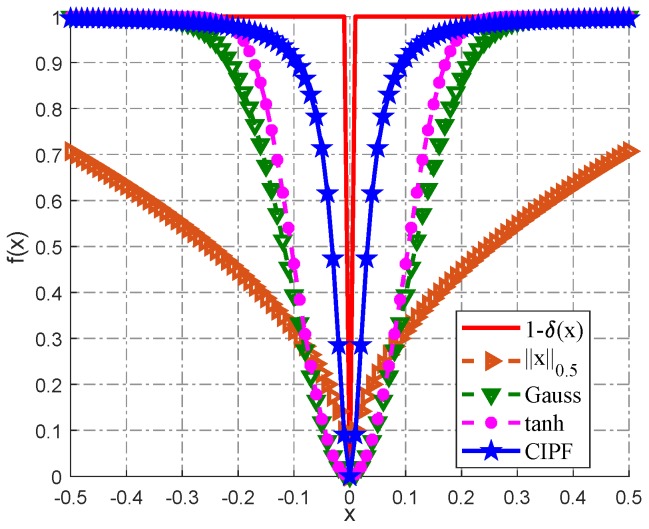
Different functions used in the literature to approximate the $L_{0}$-norm; some of them are plotted in this figure, and the $L_{0.5}$-norm is displayed for comparison. CIPF, compound inverse proportional function.

**Figure 3 sensors-18-04260-f003:**
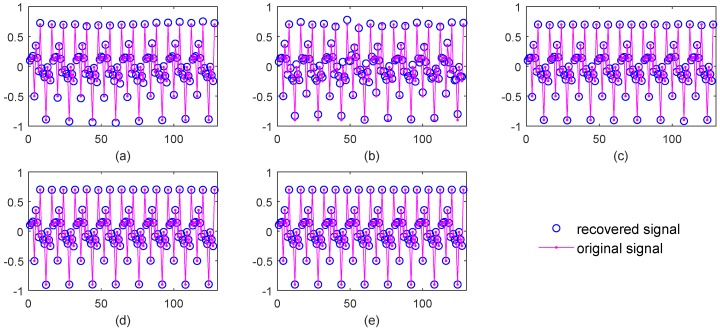
Signal recovery effect by BPDN, SL0, NSL0, L${}_{p}$-RLS, and weighted regularized smoothed $L_{0}$-norm minimization (WReSL0) when noise intensity $\delta_{N}$ = 0.2. (**a**) signal recovery by the BPDN algorithm; (**b**) signal recovery by the SL0 algorithm; (**c**) signal recovery by NSL0 algorithm; (**d**) signal recovery by the L${}_{p}$-RLS algorithm; (**e**) signal recovery by the WReSL0 algorithm.

**Figure 4 sensors-18-04260-f004:**
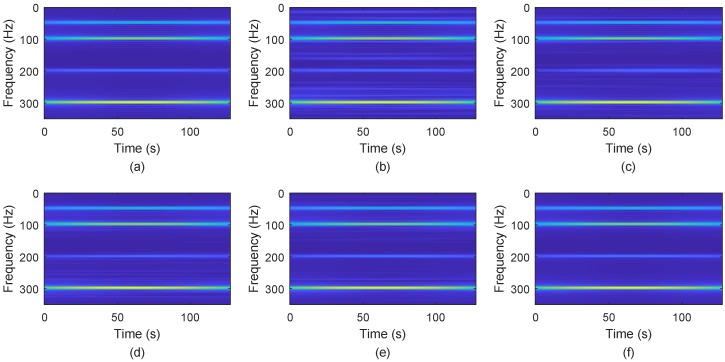
Frequency spectrum analysis of the original signal and the signal recovered by BPDN, SL0, NSL0, L${}_{p}$-RLS, and WReSL0 when noise intensity $\delta_{N}$ = 0.2. (**a**) original signal; (**b**) signal recovery by the BPDN algorithm; (**c**) signal recovery by the SL0 algorithm; (**d**) signal recovery by the NSL0 algorithm; (**e**) signal recovery by the L${}_{p}$-RLS algorithm; (**f**) signal recovery by the WReSL0 algorithm.

**Figure 5 sensors-18-04260-f005:**
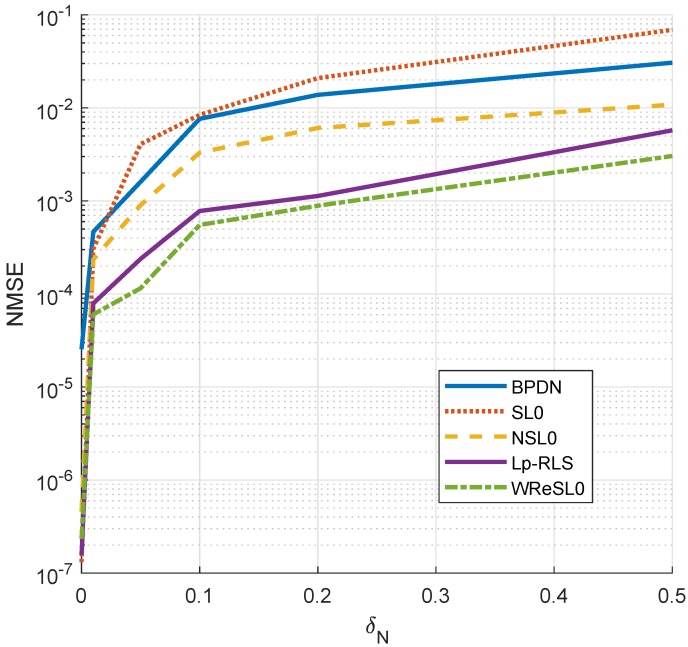
NMSE analysis by BPDN, SL0, NSL0, L${}_{p}$-RLS, and WReSL0 when noise intensity $\delta_{N}$ changes according to the sequence [0, 0.1, 0.2, 0.3, 0.4, 0.5].

**Figure 6 sensors-18-04260-f006:**
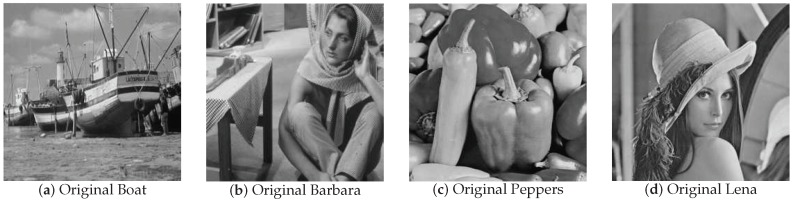
Original images: (**a**) boat; (**b**) Barbara; (**c**) peppers; (**d**) Lena.

**Figure 7 sensors-18-04260-f007:**
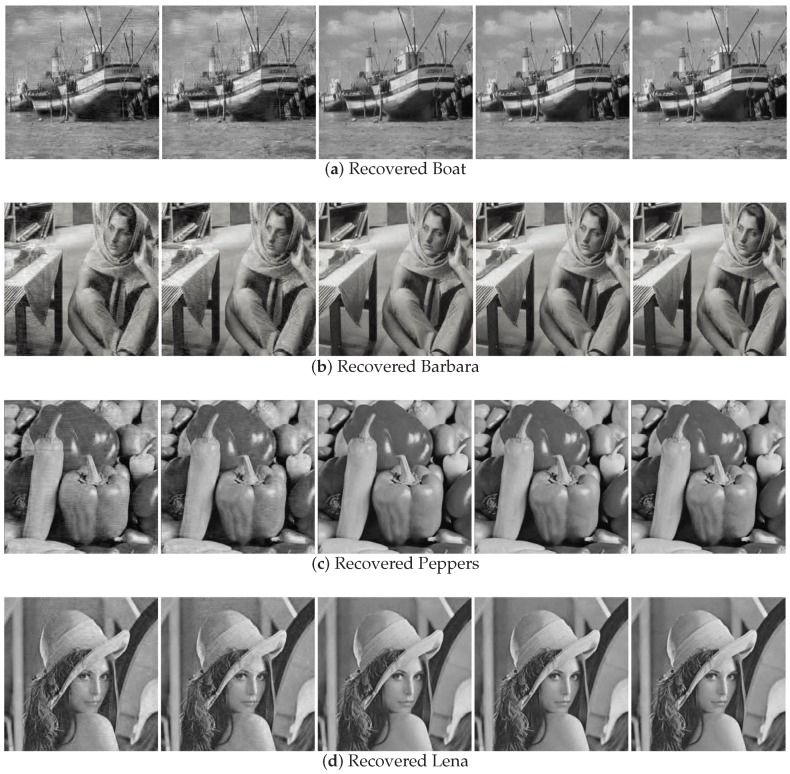
Image recovery effect by the BPDN, SL0, NSL0, L${}_{p}$-RLS, and WReSL0 algorithms with noise intensity $\delta_{N}$ = 0.01. In (**a**–**d**), from left to right, are: image recovered by the BPDN, SL0, NSL0, L${}_{p}$-RLS, and WReSL0 algorithms.

**Figure 8 sensors-18-04260-f008:**
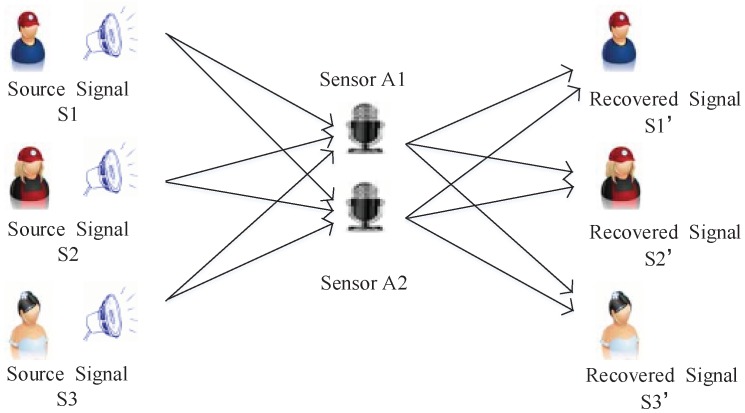
Schematic diagram of cocktail reception signal mixing.

**Figure 9 sensors-18-04260-f009:**
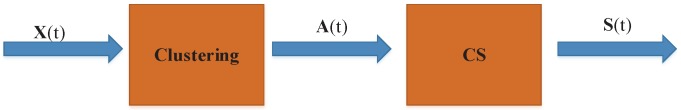
Schematic diagram of two-step method for UBSS.

**Figure 10 sensors-18-04260-f010:**
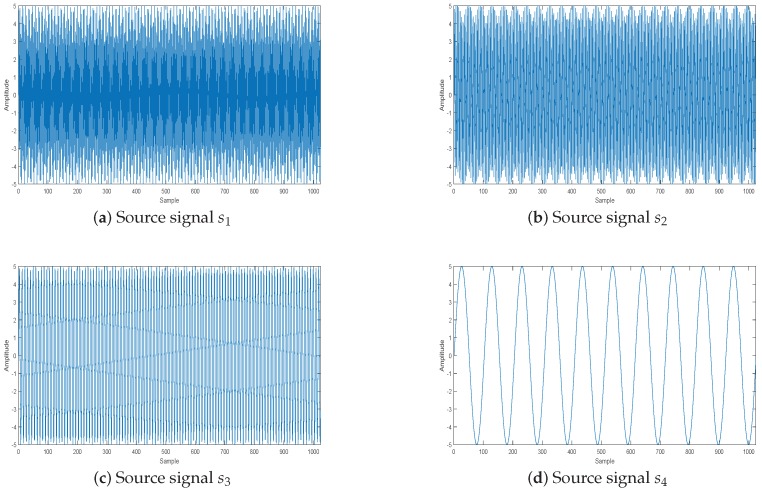
Source signal.

**Figure 11 sensors-18-04260-f011:**
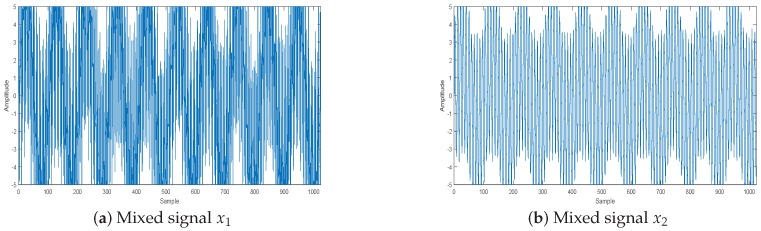
Mixed signal by sensors.

**Figure 12 sensors-18-04260-f012:**
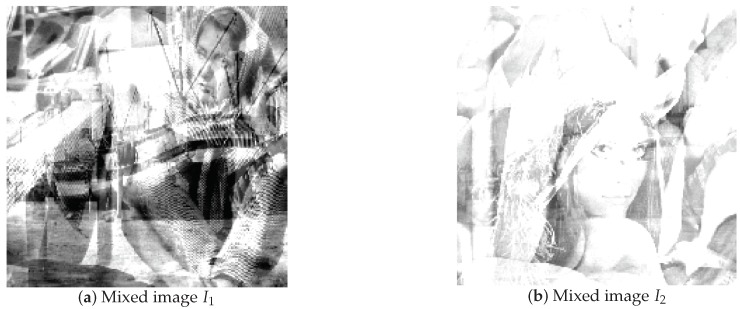
Mixed image by sensors.

**Figure 13 sensors-18-04260-f013:**
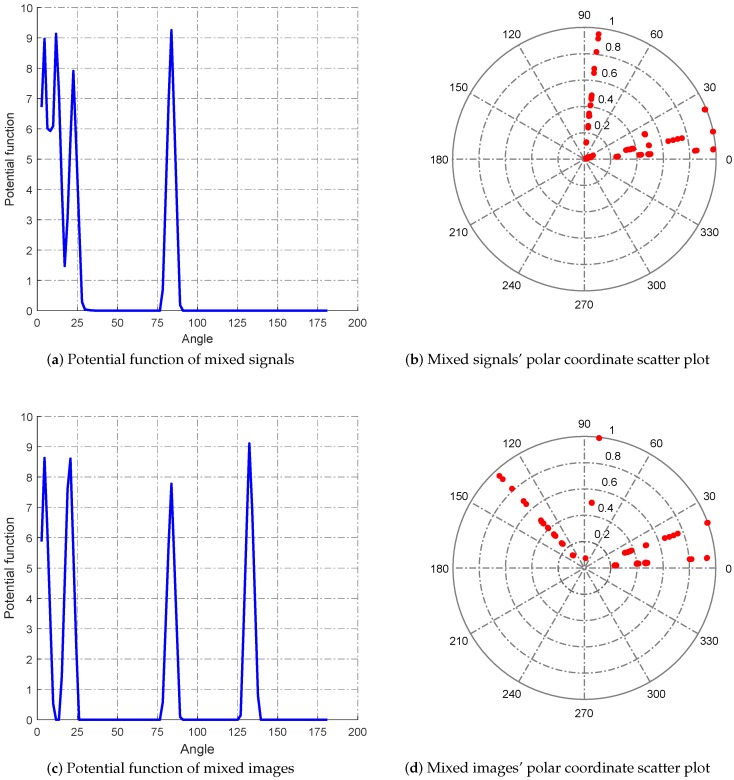
Clustering analysis.

**Figure 14 sensors-18-04260-f014:**
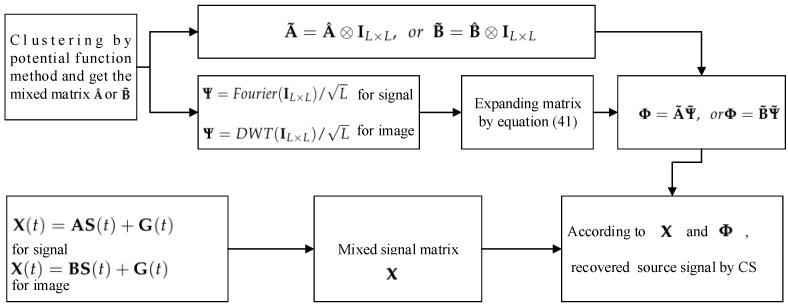
Flowchart of UBSS by CS.

**Figure 15 sensors-18-04260-f015:**
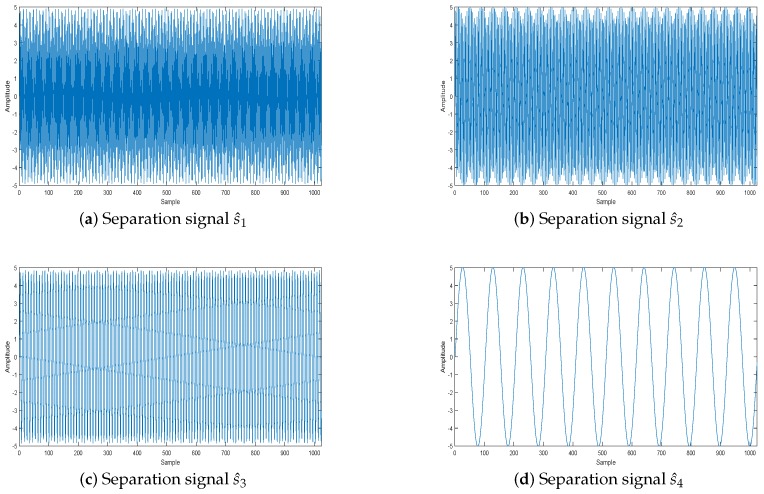
Separation signal.

**Figure 16 sensors-18-04260-f016:**
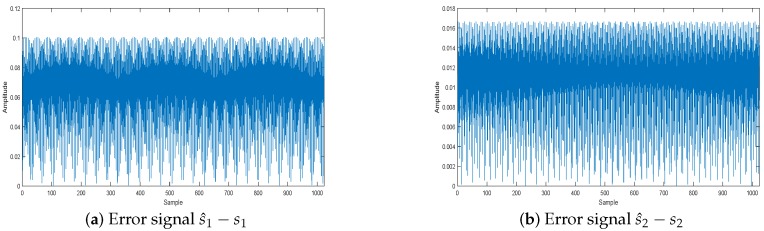
Separation signal error analysis.

**Figure 17 sensors-18-04260-f017:**
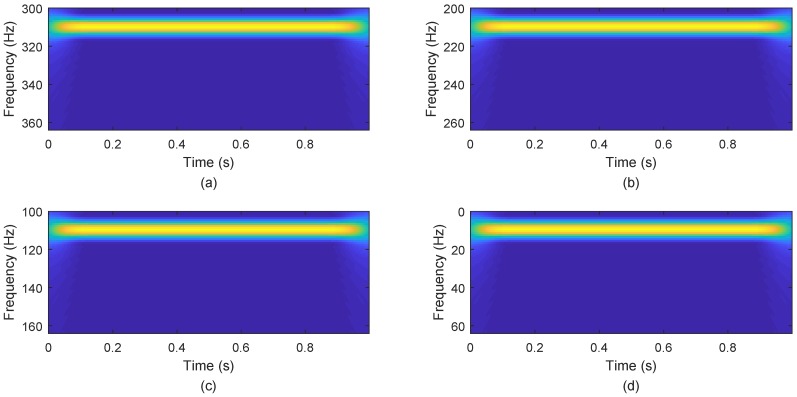
Separation signals’ frequency spectrum. Subfigures (**a**–**d**) show the frequency spectrums of separation signals ${\hat{s}}_{1}$, ${\hat{s}}_{2}$, ${\hat{s}}_{3}$, and ${\hat{s}}_{4}$.

**Figure 18 sensors-18-04260-f018:**
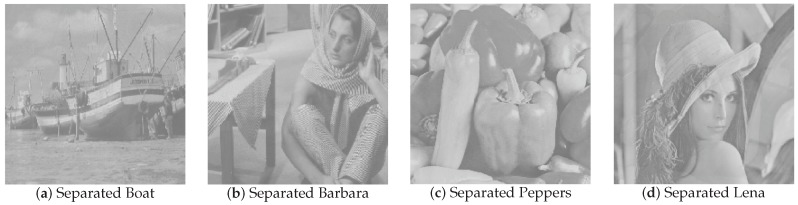
Separated images: (**a**) boat; (**b**) Barbara; (**c**) peppers; (**d**) Lena.

**Table 1 sensors-18-04260-t001:** Weighted regularized smoothed $L_{0}$-norm minimization (WReSL0) algorithm using the GD method.

● Initialization:
(1) Set $L,\mu =\sigma /\left(2\alpha \right),{\hat{x}}^{\left(0\right)}=\Phi^{H}{\left(\Phi \Phi^{H}\right)}^{-1}y$.
(2) Set $\sigma_{\max}=\sqrt{\alpha }\max\left|x\right|$, $\sigma_{\min}=0.01$, and $\sigma_{t}=\sigma_{\max}\theta^{-\gamma (t-1)}$, where $\gamma =\frac{\log_{\theta }\left(\sigma_{\max}/\sigma_{\min}\right)}{T-1}$, and *T* is the maximum number of iterations.
● **while** t<T, **do**
(1) Let $\sigma =\sigma_{t}$.
(2) Let $x={\hat{x}}^{\left(t-1\right)}$.
**for** l=1,2,…,L
(a) $x\leftarrow x-\mu {\left(e^{-}\frac{\left|x\right|}{\sigma }\right)}^{T}\frac{2\alpha \sigma^{2}x}{{\left(\alpha x^{2}+\sigma^{2}\right)}^{2}}$.
(b) $x\leftarrow x-{\left(\lambda^{-1}I_{n}+\Phi^{H}\Phi \right)}^{-1}\Phi^{H}\left(\Phi \hat{x}-y\right)$
(3) Set ${\hat{x}}^{\left(t-1\right)}=x$.
● The estimated value is $\hat{x}={\hat{x}}^{\left(t\right)}$.

**Table 2 sensors-18-04260-t002:** Signal CPU running time (CRT) analysis for BPDN, SL0, NSL0, L${}_{p}$-RLS, and the proposed WReSL0 with signal length changes according to the sequence [170, 220, 270, 320, 370, 420, 470, 520] when $\delta_{N}=0.2$.

Signal Length (n)	CPU Running Time (Seconds)				
	BPDN	SL0	NSL0	Lp-RLS	WReSL0
170	0.195	0.057	0.091	0.194	0.063
220	0.289	0.139	0.230	0.350	0.142
270	0.495	0.229	0.426	0.505	0.291
320	0.767	0.320	0.639	0.712	0.509
370	1.059	0.456	0.926	0.982	0.892
420	1.477	0.613	1.133	1.491	1.017
470	1.941	0.796	1.478	2.118	1.344
520	2.619	1.038	2.089	2.910	1.882

**Table 3 sensors-18-04260-t003:** PSNR and SSIM analysis of recovered images (boat and Barbara) by SL0, BPDN, NSL0, L${}_{p}$-RLS, and WReSL0.

Items	Barbara		Boat	
	PSNR (dB)	SSIM	PSNR (dB)	SSIM
SL0	27.983	0.981	26.959	0.969
BPDN	28.834	0.984	27.376	0.971
NSL0	31.296	0.991	31.247	0.988
L${}_{p}$-RLS	31.786	0.992	31.797	0.989
WReSL0	32.244	0.993	32.369	0.991

**Table 4 sensors-18-04260-t004:** PSNR and SSIM analysis of recovered images (peppers and Lena) by SL0, BPDN, NSL0, L${}_{p}$-RLS, and WReSL0.

Items	Peppers		Lena	
	PSNR (dB)	SSIM	PSNR (dB)	SSIM
SL0	28.677	0.982	30.334	0.987
BPDN	29.542	0.985	29.875	0.983
NSL0	31.373	0.991	32.639	0.993
L${}_{p}$-RLS	33.757	0.994	34.051	0.995
WReSL0	34.231	0.996	34.653	0.997

**Table 5 sensors-18-04260-t005:** Average SNR (ASNR) analysis for separated signals by SPM, SL0, NSL0, L${}_{p}$-RLS, and the proposed WReSL0 with $\delta_{N}$ changing according to sequence [0,0.1,0.15,0.18,0.2] with 100 runs.

Oise Intensity ($\delta_{N}$)	Error of $\hat{A}$(%)	ASNR (dB)				
		SPM	SL0	NSL0	L_*p*_-RLS	WReSL0
0	1.763	45.443	41.576	42.324	38.412	39.993
0.1	1.763	36.788	35.278	36.034	37.091	39.295
0.15	1.763	31.407	30.754	32.930	35.332	38.975
0.18	112.6	26.355	24.063	25.437	28.305	26.650
0.2	126.3	11.201	9.974	12.358	17.549	15.581

**Table 6 sensors-18-04260-t006:** APSNR analysis for separated images by SPM, SL0, NSL0, L${}_{p}$-RLS, and the proposed WReSL0 with $\delta_{N}$ changing according to the sequence [0,0.1,0.15,0.18,0.2] with 100 runs.

Noise Intensity ($\delta_{N}$)	Error of $\hat{B}$(%)	APSNR (dB)				
		SPM	SL0	NSL0	L_*p*_-RLS	WReSL0
0	3.64	16.447	19.211	20.035	16.372	18.483
0.1	3.64	15.639	16.305	17.327	15.407	17.849
0.15	3.64	13.407	14.754	14.930	14.932	17.351
0.18	133.2	9.355	11.063	11.437	10.305	11.650
0.2	142.4	5.201	5.974	6.358	3.549	5.581

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
