# Peer review of "A Regularized Weighted Smoothed *L*_0_ Norm Minimization Method for Underdetermined Blind Source Separation"

_sensors, 2018, doi:10.3390/s18124260_

Round 1
Reviewer 1 Report
Report attached.

Reviewer 2 Report
This paper presents a sparse recovery algorithm for compressed sensing, with application to underdetermined blind source separation. The authors find a better function (CIPF) to approximate L0-norm, and provide a new weighting function (exp(-|x|/sigma)), which helps promoting sparsity. In addition, gradient descent algorithm is adopted to solve the convex optimization problem and yields better recovery performance in a variety of numerical experiments. The outline is clear and the overall writing is good.
Some minor remarks on the paper.
1. Regarding CIPF parameters, what is the value of sigma? And why alpha is set to 10? Please elaborate more on this.
2. Paragraph under Eq. 12, what does factor 'c' stand for? Should it be 'alpha'?
3. In Candes' and Pant's papers, they came up with 1/(|x|+e) due to the fact that log(|x|+e) is a good approximate to L0-norm. In addition, Candes et al., addressed the 'computation on computer' problem for iterative reweighting by introducing e, although its value is sort of empirical. For the proposed weighted function in Eq. 14, what is the mathematical explanation for your choice?
4. Line 104, missing eq. reference
5. Line 112, i.i.e, is it i.i.d? or i.e.?
6. What is in Fig. 4(f)?
7. For numerical experiments, I did not see the point why 2D image recovery is added since UBSS is the target application.
8. Please add references to the selected algorithms for comparison: SL0, NSL0, Lp-RLS, etc.,.
Reviewer 3 Report
The paper proposed a method to compress and recover signal in sparse space. Comprehensive experiments are performed to demonstrate the efficacy of the proposed approach including experiments in various applications.
However, there are limited number of setting in each application. For example, in image recovery application, only 2 images are used to perform an experiment. Also, in underdetermined blind source separation, which is the application mentioned in the title, only simulated signal is used to perform an experiment. Overall, more experiments on real world data on this topic is needed to ensure applicability of the proposed approach.
Round 2
Reviewer 1 Report
Report attached.

Round 3
Reviewer 1 Report
Thank you for your explanation and further update. I now recommend acceptance and publication.